# Genome-Wide Identification, Expression, and Molecular Characterization of the *CONSTANS-like* Gene Family in Seven Orchid Species

**DOI:** 10.3390/ijms242316825

**Published:** 2023-11-27

**Authors:** Yonglu Wei, Jianpeng Jin, Zengyu Lin, Chuqiao Lu, Jie Gao, Jie Li, Qi Xie, Wei Zhu, Genfa Zhu, Fengxi Yang

**Affiliations:** Guangdong Key Laboratory of Ornamental Plant Germplasm Innovation and Utilization, Environmental Horticulture Research Institute, Guangdong Academy of Agricultural Sciences, Guangzhou 510640, China; weiyonglu@gdaas.cn (Y.W.); jinjianpeng@gdaas.cn (J.J.); zengyu_lin00@163.com (Z.L.); luchuqiao@gdaas.cn (C.L.); gaojie@gdaas.cn (J.G.); lijie@gdaas.cn (J.L.); xieqi@gdaas.cn (Q.X.); zhuwei0923@126.com (W.Z.); zhugenfa@gdaas.cn (G.Z.)

**Keywords:** *Cymbidium sinense*, *Dendrobium*, *Phalaenopsis*

## Abstract

The orchid is one of the most distinctive and highly valued flowering plants. Nevertheless, the *CONSTANS-like* (*COL*) gene family plays significant roles in the control of flowering, and its functions in Orchidaceae have been minimally explored. This research identified 68 potential *COL* genes within seven orchids’ complete genome, divided into three groups (groups I, II, and III) via a phylogenetic tree. The modeled three-dimensional structure and the conserved domains exhibited a high degree of similarity among the orchid *COL* proteins. The selection pressure analysis showed that all orchid COLs suffered a strong purifying selection. Furthermore, the orchid *COL* genes exhibited functional and structural heterogeneity in terms of collinearity, gene structure, cis-acting elements within their promoters, and expression patterns. Moreover, we identified 50 genes in orchids with a homology to those involved in the *COL* transcriptional regulatory network in Arabidopsis. Additionally, the first overexpression of *CsiCOL05* and *CsiCOL09* in *Cymbidium sinense* protoplasts suggests that they may antagonize the regulation of flowering time and gynostemium development. Our study will undoubtedly provide new resources, ideas, and values for the modern breeding of orchids and other plants.

## 1. Introduction

The orchid family (Orchidaceae) is one of the most species-rich plant families. There are approximately 30,000 orchid species with colorful and aromatic flowers that are appreciated worldwide [1,2,3]. In addition to its economic importance, the orchid offers ecological, ornamental, medicinal, aesthetic, and cultural value [4,5,6]. Molecular phylogenetic analyses suggest that the orchid family consists of five mono-phyletic subfamilies: Apostasioideae, Vanilloideae, Cypripedioideae, Orchidoideae, and Epidendroideae [2,7]. The diverse *Vanilla* genus, which belongs to the Vanilloideae family, has extensive usage in various industries, such as food, pharmaceuticals, cosmetics, beverages, and traditional crafts [8,9,10]. Epidendroideae, with approximately 21,160 species, is the largest of the five orchid subfamilies [11]. Due to their diversity, many species from several genera are very prominent industrial commodities. For instance, *Dendrobium* is a valuable traditional herb with high commercial worth [12]. *Cymbidium*, on the other hand, has been grown in China for thousands of years and is renowned as the ‘King of Fragrance’ [13].

The induction of flowering is a key step leading to proper flower development in the orchid, and several functional pathways, including photoperiod, vernalization, environmental temperature, phytohormones, and autonomous flowering pathways, have been identified to control flowering induction [14,15,16,17,18,19]. *CONSTANS-LIKE* (*COL*) genes, phosphatidyl ethanolamine-binding protein (PEBP) genes, and several members of the MADS-box gene family in these pathways have been suggested to play a key role in flowering time regulation [20,21,22]. However, due to several major technical obstacles, including the long vegetative phase, the low efficiency of the genetic transformation systems, and the time-consuming process of tissue culture, the molecular mechanism has yet to be elucidated [23].

*CONSTANS*, a member of the zinc fingered transcription factor family, is an important regulator of plant responses to the photoperiod, playing a key role in the regulation of flowering [24]. The *COL* family genes contain two conserved elements: the N-terminal BBX (B-box) domain, which consists of four cysteines with a specific structure of (C-X_2_-C-X_16_-C-X_2_-C); and the C-terminal CCT (*CONSTANS*, *CO-like*, and *TIMING of CAB1*) domain [25,26]. Since the discovery of the first *COL* gene in Arabidopsis [27], numerous members of the *COL* gene family have been identified in various plant species, including 12 in grapevine (*Vitis vinifera*) [28], 13 in lotus (*Nelumbo nucifera*) [29], 14 in Populus [30], 15 in petunia (*Petunia axillaris*) [31], 16 in rice (*Oryza sativa*) [32], 17 in Arabidopsis [20], 19 in maize (*Zea mays*) [33], 22 in sunflower (*Helianthus annuus*) [34], 25 in banana (*Musa acuminata*) [35], 26 in soybean (*Glycine max*) [36], and 42 in cotton (*Gossypium hirsutum*) [37].

The sequence of the orchid genome provides genetic resources for gene functional studies, and the study of the orchid flowering-time genes can therefore provide essential information for the further modification of orchid varieties to increase yield. Previously, there was limited information on the functions of genes involved in flowering time regulation in orchids [5,6]. Here, we identified seven *COL* genes and investigated their properties, such as chromosome location, gene organization, cis-acting elements, protein–protein interactions (PPIs), and gene expression pattern. In addition, we were surprised to find that the *CsiCOL* genes in *C. sinense* may be related to the development of the gynostemium, which is one of the typical characteristics of orchids that distinguishes them from other plants [38]. First, the overexpression of *CsiCOL05* and *CsiCOL09* in *C. sinense* protoplasts confirms the regulation of the flowering time and flower development genes. Our results provide useful information for characterizing *COL* gene functions in orchids and other plants.

## 2. Results

### 2.1. Basic Characterization of COL Genes in Orchidaceae

In the present study, the genomes of five Epidendroideae (*Cymbidium sinense*; *Csi*, *Dendrobium catenatum*; *Dca*, *Dendrobium chrysotoxum*; *Dch*, *Dendrobium huoshanense*; and *Dhu*, *Phalaenopsis equestri*; *Peq*); one Apostasioideae (*Apostasia shenzhenica*; *Ash*); and one Vanilloideae (*Vanilla planifolia*; *Vpl*). Orchidaceae species were thoroughly scanned for gene identification. Finally, a total of 68 putative *COL* genes were identified in all seven orchid families by the HMM program and subsequently verified by the Pfam and blastp databases, with all *COL* genes containing both B-box and CCT domains. To distinguish the sixty-eight genes, we named them *CsiCOL1* to *CsiCOL10*, *DcaCOL01* to *DcaCOL09*, *DchCOL01* to *DchCOL10*, *DhuCOL01* to *DhuCOL08*, *PeqCOL01* to *PeqCOL07*, *AshCOL01* to *AshCOL10*, and *VplCOL01* to *VplCOL14*, according to their physical location on the chromosomes. Detailed information is provided in Appendix A. The *DhuCOL* genes were dispersed on six chromosomes of *D. huoshanense*: one in chromosome 1 (*DhuCOL05*), 3 (*DhuCOL04*), 14 (*DhuCOL03*), and 18 (*DhuCOL06*); and two in chromosome 7 (*DhuCOL07* and *DhuCOL08*) and 17 (*DhuCOL01* and *CsiCOL02*), respectively. The *VplCOL* genes were scattered on nine chromosomes and one contig of the *V. planifolia* genome by one in chromosome 3 (*VplCOL12*), 4 (*VplCOL11*), 7 (*VplCOL08*), 8 (*VplCOL07*), 9 (*VplCOL06*), and JADCNL010000338 (*VplCOL01*); and two in chromosome 2 (*VplCOL13* and *VplCOL14*), 5 (*VplCOL09* and *VplCOL10*), 13 (*VplCOL04* and *VplCOL05*), and 14 (*VplCOL02* and *VplCOL03*), respectively. The *DchCOL* genes were scattered on nine chromosomes and one contig of the *D. chrysotoxum* genome by one in chromosome 7 (*DchCOL06*), 11 (*DchCOL03*), 16 (*DchCOL02*), 18 (*DchCOL09*), and unchr_scaffold_742 (*DchCOL01*); and two in chromosome 9 (*DchCOL07* and *DchCOL08*) and 12 (*DchCOL04* and *DchCOL05*), respectively. The *CsiCOL* genes were distributed on seven chromosomes and one contig of the *C. sinense* genome, with one in chromosome 3 (*CsiCOL04*), 7 (*CsiCOL10*), 8 (*CsiCOL09*), 15 (*CsiCOL07*), 20 (*CsiCOL03*), and contig4269 (*CsiCOL08*); and two in chromosome 6 (*CsiCOL05* and *CsiCOL06*) and 9 (*CsiCOL01* and *CsiCOL02*), respectively.

The amino acid sequences of seven orchid *COL* proteins range from 163 (*Ash-COL01*) to 499 (*DhuCOL01*) in length, and the molecular weights are between 18.43 kDa (*AshCOL4*) and 54.68 kDa (*DhuCOL01*). The minimum and maximum isoelectric points are 4.86 (*VplCOL13*) and 9.36 (*VplCOL02*), respectively, among the orchid COL genes. The protein instability index analysis showed that, except for *AshCOL01* (39.52), *DcaCOL01* (29.5), and *VplCOL06* (36.52), all seven Orchidaceae *COL* members belong to unstable proteins (instability index > 40) [39]. The prediction results of the subcellular location showed that the *COL* genes of Orchidaceae are localized in the nucleus, mitochondria, and several other locations.

### 2.2. Phylogenetic Analysis of COL Genes

In order to understand the evolutionary relationships of *COL* family genes in Orchidaceae, we constructed an unrooted tree (Figure 1), using 101 *COL* proteins from *A. thaliana* (17), *O. sativa* (16), *A. shenzhenica* (10), *D. catenatum* (9), *D. huoshanense* (8), *V. planifolia* (14), *D. chrysotoxum* (10), *C. sinense* (10), and *P. equestri* (7). All of the *COL* proteins were verified by the Pfam and Blastp databases as containing both the B-box and the CCT domains. In Arabidopsis, *COLs* are divided into three groups according to their sequence alignment, with group I *COLs* containing two B boxes, group II *COLs* containing one normal and one divergent B box, and group III *COLs* containing only one B box domain [20,32]. Each group contained at least one *COL* protein from the nine different plant species, whereas *P. equestri* was not expected in III. The distribution of Orchidaceae COL proteins was twenty-eight (*AshCOL02*-*AshCOL04*, *AshCOL07*, *AshCOL09*, *DcaCOL02*, *DcaCOL09*, *DhuCOL02*, *DhuCOL03*, *DhuCOL06*, *DhuCOL07*, *VplCOL01*, *VplCOL04*-*VplCOL06*, *VplCOL08*, *VplCOL13*, *DchCOL03*, *DchCOL05*, *DchCOL07*, *DchCOL09*, *CsiCOL01*, *CsiCOL04*, *CsiCOL05*, *CsiCOL09,* and *PeqCOL02*-*PeqCOL04*) in group I; twenty-eight (*Ash-COL01*, *AshCOL06*, *AshCOL08*, *AshCOL10*, *DcaCOL01*, *DcaCOL03*, *DcaCOL04*, *DcaCOL06*, *DcaCOL07*, *DhuCOL01*, *DhuCOL08*, *VplCOL02*, *VplCOL03*, *VplCOL09*, *VplCOL10*-*VplCOL12*, *DchCOL04*, *DchCOL08*, *DchCOL10*, *CsiCOL02*, *CsiCOL03*, *CsiCOL06*, *CsiCOL08*, *PeqCOL01*, and *PeqCOL05*-*PeqCOL07*) in group II; and twelve (*AshCOL05*, *DcaCOL05*, *DcaCOL08*, *DhuCOL04*, *DhuCOL05*, *VplCOL07*, *VplCOL14*, *DchCOL01*, *DchCOL02*, *DchCOL06*, *CsiCOL07*, and *CsiCOL10*) in group III.

### 2.3. Sequence Structure Analysis of COL Members

Sequence structure information was predicted using MEME. Fifteen types of conserved motifs were represented by numbers from 1 to 10 (Figure 2A and Appendix A). All Orchidaceae *COL* members contain two conserved domains, one or two N-terminal B-boxes (motif 2 and motif 5), and one CCT domain (motif 1) near the C-terminus. The number of *COL* motifs ranged from two (*VplCOL01*) to seven (*CsiCOL06*). Even though the majority of the orchid *COL* proteins contained these six conserved protein domains, the motifs of each sub-clade are still different from each other. For instance, motifs 6 and 8 were present only in class II. An orchid VP motif (motif 9) alongside the CCT domain, which is important for binding to the *COP1* gene to regulate light signaling cascades [40]. The conserved motif was identified in the group I *COL* members in Orchidaceae. It has also been detected in *A. thaliana* [26] and cucumber [41].

The intron–exon structure was analyzed (Figure 2B) to gain further insight into the characteristics of orchid *COL* genes. The results showed that the orchid *COL* family is composed of one to six exons. Although the gene structure is similar in each subgroup, orchid *COL* genes have a high degree of variation in intron length and exon number. Although most of the orchid *COL* genes in group II have longer introns than the other subgroups, the gene structure of *VplCOL13* showed a significant difference in the length of the introns (Figure 2B), which may be a peculiar feature of orchids.

### 2.4. Chromosomal Location, Collinearity and Evolutionary Analysis of Orchid COL Genes

According to the annotation file of the four orchid species, the densities of the 300 kb inheritance interval genes were obtained and then further transformed into a gradient colored heatmap on the orchid chromosome or scaffold. The Tbtools program was used to visualize the chromosomal locations of the *COL* genes [42] in the chromosomal genome of the four orchids. The 39 *COL* genes are distributed unevenly and widely over the 28 chromosomes (Figure 3 and Appendix A). There are 9, 9, 8, and 13 *COL* genes in the *C. sinense*, *D. chrysotoxum*, *D. huoshanense,* and *V. planifolia* genomes, respectively (Figure 3A–D). Interestingly, only one or two of the *COL* genes were observed on the same chromosome in four species of Orchidaceae; otherwise, each *COL* gene in group III of *C. sinense*, *D. chrysotoxum,* and *D. huoshanense* was located on an independent chromosome. However, four chromosomes (Chr3, 4, 5, and 14) were exclusively divided into six COLs belonging to group II in the genome of *V. planifolia*.

A collinearity analysis was performed between *C. sinense* and five plant species (Appendix A), including one dicot (*A. thaliana*) and four monocots (*D. huoshanense*, *D. chrysotoxum, V. planifolia*, and *O. sativa*). The number of *COL* collinearity gene pairs between *C. sinense* and *A. thaliana*, *D. huoshanense*, *D. chrysotoxum*, *V. planifolia,* and *O. sativa* are 1, 7, 8, 8, and 2, respectively (Appendix A). The comparative genome analysis revealed more conserved collinear blocks (Figure 4), and the collinear analysis with orchids revealed multiple-to-one phenomena. For example, *CsiCOL1* has a collinearity relationship with both *VplCOL06* and *VplCOL08*. However, relatively few collinearity gene pairs were detected between *C. sinense* and the model plant *A. thaliana* or *O. sativa*. With more collinearity gene pairs of the *COL* genes (one on one), the collinearity analysis between *C. sinense* and the other three Orchidaceae showed that *COL* genes in groups I, II, and III were involved in the formation of collinearity gene pairs; however, only group I and II COL members in *C. sinense* produced collinearity with those in *A. thaliana* and *O. sativa*, indicating that the *COL* sequences in group I and II are relatively conserved in the evolutionary history.

For the analysis of the selection pressure, we calculated the ratio of the non-synonymous substitutions per non-synonymous site (Ka) to the synonymous substitutions per synonymous site (Ks) of 40 gene pairs that were selected on the basis of sequence similarity. The results showed that the Ka/Ks ratios of all *COL* genes were <1, with most values < 0.4, indicating that all orchid *COLs* have undergone strong purifying selection (Appendix A) [43].

### 2.5. Cis-Acting Elements in the Promoters of Orchid COL Genes

Cis-acting elements in the promoters of orchid *COL* genes: In the promoter regions of Orchidaceae *COL* genes (Figure 5 and Appendix A). the composition of cis-acting elements was detected. We detected 941 light-responsive evolved elements, including G-box, GT1 motif, GATA motif, ACE, and 3-AF1 binding sites. These elements indicate that *COL* genes can be used as light sensors in flowering plants [44]. Among them, 70 and 209 promoter regions were found to contain GT1 motif and G-box cis-acting elements, respectively. Phytohormone-responsive elements, mainly correlated with GA (GARE motif), MeJA (CGTCA motif), and auxin (AuxRR core), were also detected in the promoter regions. Moreover, the developmental element CAT-box was found at three promoter regions, and its specific function is associated with meristematic expression. A circadian element was found in 3 (*AshCOL01*, *AshCOL06,* and *AshCOL09*) in *A. shenzhenica*, 2 (*CsiCOL01* and *CsiCOL03*) in *C. sinense*, 2 (*DcaCOL05* and *DcaCOL09*) in *D. catenatum*, 1 (*DhuCOL07*) in *D. huoshanense*, and 1 (*VplCOL06*) in *V. planifolia*, corresponding to the circadian expression pattern of *COL* genes [45,46]. The possible functions and expression patterns of *COL* genes are related to the composition of cis-acting elements.

### 2.6. Expression Profiles of Orchid COL Genes in Different Tissues and Differentially Expressed CsiCOL Genes during ABA Treatment

We analyzed the expression based on seven Orchidaceae transcriptome data in different tissues, including roots, stems, leaves, flowers, fruits, and seeds, to shed light on the potential functions of orchid *COL* genes during plant development. The *COL* genes showed different patterns of expression in the different tissues (Figure 6). For example, *VplCOL08* was highly expressed in all tissues, while *VplCOL01* and *VplCOL09* were lowly expressed. All genes were highly expressed in specific tissues. This suggests that they may have critical functions in these tissues. For example, *CsiCOL4* and *CsiCOL6* showed a high level of expression in flowers but a low level of expression in leaves. Interestingly, *CsiCOL* genes showed tissue-specific expression in individual floral organs of different flower varieties (Figure 7). For example, *CsiCOL09* was more highly expressed in the column, while *CsiCOL01* showed complementary expression patterns, suggesting that the two genes play opposite roles in column formation; similar patterns occur in regard to the Arabidopsis flowering time, which is regulated by *CO* and *COL09* genes [47]. In addition, *CsiCOL05* was highly expressed in all cultivars, indicating an essential function in flower development. Under ABA treatment, the expression of *CsiCOL04* and *CsiCOL10* in the leaves was highly reduced (Appendix A).

### 2.7. Protein Structure Prediction

The tertiary structure of most orchid *COLs* is highly conserved, characterized by four β-sheet and four α-helices; the N-terminus had an anti-parallel β-sheet formed by β1 and β2. The α1 helix aligned in parallel with the β-sheet. A C-terminal short α2 helix was present vertically against α1, forming the compact structure (Figure 8). The mirrored structure with a similar arrangement was connected by a loop structure. Except for 3 *COLs* (*VplCOL02*, *VplCOL03,* and *CsiCOL04*), which showed two β-sheets and three α-helices, 16 *COLs* (*AshCOL02*, *AshCOL05*, *AshCOL08*, *DcaCOL05*, *DcaCOL08*, *DhuCOL04-07*, *VplCOL14*, *DchCOL01*, *DchCOL02*, *DchCOL05-07*, *CsiCOL07*, and *CsiCOL10*) showed two β-sheets and two α-helices, *DchCOL09* had two β-sheets and one α-helices, 2 COLs (*VplCOL05* and *VplCOL06*) had two α-helices, and 3 *COLs* (*AshCOL07*, *VplCOL02*, and *PeqCOL04*) exhibited only two β-sheets. Each strand in the asymmetric unit interacts with symmetrically related molecules on both sides to form an extended linear head-to-tail oligomeric configuration, resulting in an enhanced performance in transcriptional regulation [48]. Secondary structure prediction revealed that all orchid *COL* proteins are composed of an α-helix (Hh), extended strands (Ee), β-turns (Tt), and random coils (Cc), which, on average, account for 31.07%, 10.65%, 3.14%, and 55.15% of the protein structure, respectively (Appendix A).

### 2.8. Correlation Analysis between CsiCOLs and Regulation of Co-Activity-Related Transcription Factors

To determine the potential transcriptional regulation mechanism of *CsiCOL*, the cor-relation analysis between the expression of the *CsiCOLs* and CO-activated transcription factor was performed (Figure 9). The results indicated that three groups of *CsiCOL* members showed a significant correlation with SHAGGY-like kinase 12 (*SK12*: *Mol003238*, *Mol012561*, and *Mol003091*) and PSEUDO-RESPONSE REGULATOR (*PRR*: *Mol011035* and *Mol029571*) transcription factors, suggesting conserved functions of the genes within each cluster, such as, in plant growth, stress-related metabolic regulation [49], and the circadian clock [50]. JUMANJI 28 (JMJ28: *Mol011862*) was significantly related to *CsiCOLs* in group I and group II. FLOWERING BHLH (*FBH*: *Mol010847*) was significantly correlated with *CsiCOLs* only in group I, while FK506-binding protein (FKBP12: *Mol015806*) was significantly correlated with *CsiCOLs* only in group III. Additionally, three genes were only significantly associated with *CsiCOLs* in group II, that is, GIGANTEA (GI: *Mol017612*), TARGET OF AT (TOE: *Mol008285*), and FLAVIN-BINDING KELCHREPEAT F-BOX1 (FKF1: *Mol013985*), respectively. This also indicates that the functions of *CsiCOL* genes diversified during evolution.

### 2.9. Exploration of Subcellular Localization and Regulatory Analysis by Transient Overexpression of CsiCOLs in Cymbidium Protoplasts

Protoplast transformation is particularly important for studying gene function in crop species, which often have unique genetic traits that are not present in model plants [51,52]. To elucidate the influence of *CsiCOLs* on the temporal association of flowering and floral morphogenesis (Figure 10A), we initiated a transient overexpression experiment, using *CsiCOL05* and *CsiCOL09* in *C. sinense* protoplasts. Subcellular localization studies reveal a nuclear disposition of *CsiCOL05*. In contrast, the subcellular location of *CsiCOL09* is observed in both the nucleus and cell membrane (Figure 10B). These results, derived from subcellular localization assessments, suggest that different *COLs* genes may have different functions in orchids. We selected ten candidate genes: four FLOWERING LOCUST (*FT*), TERMINAL FLOWER 1 (*TFL1*), APETALA1 (*AP1*), SUPPRESSOR OF OVEREXPRESSION OF CONSTANS1 (*SOC1*), LEAFY (*LFY*), and AGAMOUS1 (*AG1*), based on gene ontology and the extant literature, which were then validated using quantitative real-time polymerase chain reaction (qRT-PCR) (Appendix A). The *FT/TFL1* genes form the phosphatidylethanolamine binding protein (PEBP) family, which has been implicated in activating flowering downstream of *COL* [21,53,54]. Previous studies have implicated *AP1*, *SOC1*, *LFY*, and *AG1* in the regulation of flowering duration and floral patterning in orchids [6,55,56,57]. Notably, *CsiCOL05* had a significantly higher expression; meanwhile, for *CsiCOL09,* the expression level was much lower in gynostemium (Figure 10A). The relative expression level of four genes (*CsiFT*, *CsiAP1*, *CsiSOC1*, and *CsiLFY*) was significantly upregulated compared to the control, and the expression level of two genes (*CsiTFL1* and *CsiAG1*) was significantly downregulated in *CsiCOL05*-OE (Figure 10C). In contrast, the opposite result was observed for *CsiCOL09*-OE (Figure 10D). In summary, our results demonstrated that *CsiCOL05* and *CsiCOL09* have similar functions but different regulatory modes.

## 3. Discussion

Although the *COL* gene family has been reported in many plants, it has rarely been reported in Orchidaceae. The orchid is a mainstay of the worldwide floriculture trade and a popular ornamental crop [58,59], so the completion of the genomes of orchids provided an opportunity to study the *COL* gene family in Orchidaceae species. A comparison of *COL* gene families in seven orchids showed that they all have the typical conserved domains of the BBX and CCT domains [26]. The occurrence of B-box domains in orchid species may suggest that *COLs*, which respond to biotic [60] and abiotic stresses [61] in angiosperms, may control similar functions in Orchidaceae. The second motif of the CCT domain is important in mediating protein–protein interactions, such as Nuclear Factor-Y (*NFY*) [62], CONSTITUTIVE PHOTOMORPHOGENIC 1 (*COP1*) [63], and TOPLESS (*TPL*) [64]. Therefore, variations in CCT domains could potentially explain the significant diversity in flowering time observed across Orchidaceae [65].

In this study, the *COL* gene family was systematically compared and analyzed. Seven orchids, Arabidopsis, and rice were used to construct a phylogenetic tree. However, not all of the COL proteins that are clustered in group I have two B-box domains. For example, CsiCOL04 and CsiCOL09 were classified in group I but contain only one B-box domain. In this study, the phylogenetic tree-based classification results are not exactly the same as in Arabidopsis [20,32]. The phylogenetic relationship of the orchid *COL* genes in accordance with that of the Orchidaceae genome indicates the conservation nature and necessity of the COL genes in orchids [66]. During plant evolution, gene duplication events played a critical role in expanding gene families [67]. There were only two pairs in *D. huoshanense*, which was much less than in *V. planifolia* (8). The significant proliferation of orchid *COL* duplicates in *V. planifolia* may be attributed to the protracted flowering duration through natural selection. The collinearity analysis revealed that segregation was key to *COL* gene family expansion. As the first study on *COL* gene families in orchids, our study serves as a useful data resource for future comparative and functional genome studies on *COL* gene families.

Among the hundreds of flowering time genes that have been described to date [68]. *COL* is a key regulator of photoperiod flowering [69]. *COL* binds the conserved TGTG (N2-3) ATG motif in the *FT* promoter region, thereby activating *FT*/*TFL* to regulate the flowering time [70,71]. However, *FT* and *TFL1* have contrasting effects on both the flowering time and floral pattern [72]. Until now, only a few studies have defined the COL gene family in Orchidaceae species. The overexpression of *PaCOL1* in Arabidopsis exhibited earlier flowering under short-day (SD) conditions [73]. Two genes from cymbidium homology with *CsiCOL05*, *CsCOL1*, and *CeCOL* were overexpressed in Arabidopsis, resulting in early flowering and increased levels of *AtFT* expression under long-day (LD) conditions [74]. Similarly, the ectopic expression of Orchid *FT/TFL* from Oncidium Gower Ramsey and Dendrobium in Arabidopsis promoted or inhibited the flowering time and loss of inflorescence indeterminacy [75,76]. The basic leucine-zipper (*bZIP*) transcription factor FLOWERING LOCUS D (*FD*) plays a key role in the opposing interaction of FT and TFL1, which is dependent on competitive binding partners. This interaction triggers the transition from vegetative to reproductive growth by activating genes related to inflorescence and floral meristem identity, such as *SOC1*, *AP1*, and *LFY* [21,77,78]. Simultaneously, *AG1* was found to be linked to the formation of the column in orchids [57,79]. Combined with the changes of related genes after the overexpression of *CsiCOL05* and *CsiCOL09* in protoplasts, we speculated over the antagonistic effects of *CsiCOL05* and *CsiCOL09* on the regulation of the flowering time and gynostemium development in *C. sinense* (Figure 11).

In addition, in grapevine, *VvCOL1* has been implicated in regulating the induction and maintenance of bud dormancy [80]. *MaCOL1* was reported to be associated with stress tolerance and abiotic fruit ripening in the banana [81]. *TaCol-B5*, orthologous to COL5, modifies spike architecture and enhances grain yield in wheat [82]. The participation of ABA is seen in the regulation of plant growth and plant development, and it helps ensure yield stability [83]. In pak choi, the floral transition can be expedited, as ABA directly stimulates *BrCO* transcription through the involvement of *BrABF3* (abscisic acid-responsive transcription factors, *ABF*) [84]. Upon ABA treatment, *CsiCOL5* exhibited heightened expression throughout all tissues, underscoring its crucial involvement in the overall growth and development of *C. sinense*.

With the application of cell engineering and gene editing technologies in orchid molecular breeding [85,86], we anticipate the identification of a growing number of valuable functional genes. Orchid *COLs*, serving as key functional genes governing abiotic stress responses, flowering time regulation, and flower development within the Orchidaceae family, represent a valuable resource for enhancing orchid breeding. For instance, the Cymbidium gynostemium-like perianth variety will lose its floral scent [56,87], while the regulation of gynostemium development may be influenced by *CsiCOLs.* Moreover, a majority of orchids produce exceedingly minuscule seeds, often described as ‘dust-like’ [88,89]. Intriguingly, recent research has uncovered that the *COL* gene has the potential to govern seed size in plants [90,91]. This is not only essential for advancing the creation of new varieties with desirable traits through molecular breeding methods but also for comprehending the consequences of gene duplication and loss in the evolution of Orchidaceae and other plant species.

## 4. Materials and Methods

### 4.1. Basic Characterization of COL Genes in Orchidaceae

All *COL* proteins were identified as containing both B-box and CCT domains [20]. All *COL* genes in the genomes of *C. sinense*, *P. equestris*, *D. catenatum*, *D. chrysotoxum*, *D. huoshanense*, *V. planifolia*, and *A. shenzhenica* were identified using the Hidden Markov Model (HMM) program and related Pfam accessions (B-box and CCT domains corresponding to PF00643.19 and PF06203.9). All selected *COL* proteins were further identified using the Pfam database (http://pfam.xfam.org/, (accessed on 26 December 2022)) and Blastp in NCBI (https://www.ncbi.nlm.nih.gov/, (accessed on 30 December 2022)). The conserved domains B-box and CCT were confirmed. In order to distinguish the *COL* genes, we named them on the basis of their physical location on the chromosomes in the orchid’s genome. Basic information on the number of amino acids, molecular weight, theoretical isoelectric point (pI), and instability index (with a value < 40 considered as stable) was obtained using the ProtParam tool (https://web.expasy.org/protparam/, (accessed on 18 January 2023)). Then, the subcellular location information was obtained using the online tool BUSCA (http://busca.biocomp.unibo.it, (accessed on 22 January 2023)). All the basic information of the *COL* genes can be found in Appendix A.

### 4.2. Phylogenetic Tree Construction

To explore the phylogenetic relationships and taxonomy of *COL* genes, the phylogenetic tree was constructed using *COL* proteins from seven orchids, Arabidopsis, and rice. The *AthCOL* and *OsaCOL* protein sequences come from the TAIR database (http://www.arabidopsis.org/, (accessed on 15 December 2022)) and Rice Genome Annotation Project (http://rice.plantbiology.msu.edu/, (accessed on 15 December 2022)). Sequence alignments were performed using the most accurate algorithm, JTT + I + G4, with 1000 cycles of iterative refinement, using MAFFT v.7.475 [92]. The phylogenetic tree was reconstructed from 1000 replicates of the ultrafast bootstrap ML tree, using IQ-TREE v1.6.12 [93], with the selection of the best-fitting model by ModelFinder [94].

### 4.3. Chromosomal Location, Comparative Genome Collinearity Analysis, and Selective Pressure

Chromosomal location, comparative genome collinearity analysis, and selective pressure. The collinearity relationship of *COL* genes in *C. sinense* and the other five plant species (*D. huoshanense*, *D. chrysotoxum*, *V. planifolia*, *A. thaliana* and *O. sativa*) was identified using MCScanX [95]. The corresponding *COL* genes were mapped to chromosomes based on physical location from the orchid genome database, the TAIR database, and the rice genome annotation project. Gene colinear relationships were visualized using Tbtools v1.125 and Circos software v2.0 [96]. KaKs_Calculator 3.0 software [97] was then used to calculate the synonymous substitution rate (Ka) and nonsynonymous substitution rate (Ks) for gene pairs.

### 4.4. Conserved Motifs of Orchidaceae COL Genes

Basic information, such as the physical location, amino acid sequence, and nucleotide sequence, of orchid *COL* genes was obtained from seven Orchidaceae genomes. Conserved motifs were identified using the MEME online site (http://meme-suite.org/tools/meme, (accessed on 11 February 2023)) with the following parameters: maximum motifs number, 10; minimum and maximum widths, 6 and 50. The basic information about the sequence of the motifs is listed in Appendix A.

### 4.5. Analysis of the Cis-Acting Elements

For the analysis of cis-acting elements in their promoter region, the upstream sequences (2000 bp) of the orchid *COL* genes were collected. This analysis was carried out with PlantCARE (http://bioinformatics.psb.ugent.be/webtools/plantcare/html/, (accessed on 19 February 2023)) [98], and the results were exported with Tbtools [42].

### 4.6. Growth Conditions of C. sinense, Sample Collection, and Abscisic Acid (ABA) Treatments

*C. sinense* ‘Qi Hei’ is a famous traditional cultivar in China. For the ABA treatment, we used 2-year-old *C. sinense* ‘Qi Hei’ in flowerpots sprayed weekly with 100 μM ABA for one month, with deionized water treatment as control groups. Each biological replicate was a mixture of three plants. All the plants in this study were grown in the greenhouse of the Institute of Environmental Horticulture of the Guangdong Academy of Agricultural Sciences (Guangzhou, China). The greenhouse was maintained at 25 ± 1 °C and 80% humidity. Roots, stems, leaves, flowers, and fruit tissues were collected from the cultivated plantlets. All the samples were collected and immediately frozen in liquid nitrogen and then stored at −80 °C until they were used for further analysis.

### 4.7. Protein Tertiary Structure Prediction

SWISS-MODEL (https://swissmodel.expasy.org/interactive, (accessed on 25 February 2023)) [99] was used to draw and visualize the protein tertiary structure prediction of orchid *COLs*. The tertiary structure is rainbow-colored to represent order from the N-terminus to C-terminus. The program SOPMA (https://npsa-prabi.ibcp.fr/, (accessed on 25 February 2023)) [100] was used to predict the secondary structure.

### 4.8. Correlation Analysis between CsiCOLs and Transcription Factors

To predict the regulation of *CsiCOLs* activity, transcription factors whose function has been verified in Arabidopsis [101] were used as query sequences to identify the homologous genes of *C. sinense*, including TEOSINTE BRANCHED1/CYCLOIDEA/PROLIFERATINGCELLFACTOR (*TCP*) (*AtTCP2*, *AtTCP3*, *AtTCP4*, *AtTCP10* and *AtTCP24*), PHYTOCHROME AND FLOWERING TIME1 (*PFT1*), *GI*, *JMJ28*, *FBH1*, *FBH2*, *FBH3*, *FBH4*, CYCLING DOF FACTOR (*CDF*) (*CDF1*, *CDF2*, *CDF3*, *CDF4*, *CDF5* and *CDF6*), *TOE1*, *TOE2*, *TOE3*, CRYPTOCHROME (*CRY*) (*CRY1* and *CRY2*), PHYTOCHROME-DEPENDENT LATE-FLOWERING (*PHL*), FKF1, PHYTOCHROME (*PHY*), *PHYA*, *PHYB*, HIGH EXPRESSION OF OSMOTICALLY RESPONSIVE GENE 1 (*HOS1*), TIMING OF CAB EXPRESSION 1 (*TOC1*), PRR (*PRR5*, *PRR7* and *PRR9*), CONSTITUTIVE PHOTOMORPHOGENIC 1 (*COP1*), SUPPRESSOR OF PHYTOCHROME A (*SPA*) (*SPA1*, *SPA3* and *SPA4*), *SK12*, *FKBP12,* and EARLY FLOWERING 3 (*ELF3*) (Appendix A). In addition, the ggcor function in R language 4.2 was used to visualize the correlation heat map.

### 4.9. C. sinense Leaf Protoplast Transformation and Subcellular Localization

Protoplast isolation and transfection from the leaf bases of *C. sinense* were carried out based on previous protocols [102,103]. The vectors used were generated by cloning the coding sequences (CDSs) of *CsiCOL05* and *CsCOL09* into the PAN580-GFP vector. The aforementioned vectors containing pAN580-GFP, pAN580-*CsiCOL05*-GFP, and pAN580-*CsCOL09*-GFP were then introduced into *Escherichia coli* DH5α competent cells (Tiangen, Beijing, China) according to the manufacturer’s guidelines. This step was followed by large-scale bacterial propagation and extraction of plasmid DNA, using the Endo-Free Plasmid Maxi Kit (Omega Biotek, Norcross, GA, USA). The resulting plasmid DNA was then concentrated and adjusted to various concentrations, up to 2.0 μg/μL, before being introduced into orchid protoplasts. After incubation at 23 °C in a light-free environment for 16 h, the fluorescence emitted by the GFP or GFP-protein fusions could be observed using an LSM710 confocal laser scanning microscope. For nuclear visualization, the transfected protoplasts were stained with 50 μg/mL DAPI (Sigma-Aldrich Chemie, Steinheim, Germany) at 37 °C for 10 min. Excitation of DAPI signals was performed with a blue diode laser, using a 405 nm excitation line with a 485 nm long-pass barrier filter.

### 4.10. RNA Extraction and qRT-PCR Analysis

The transfectants (5 × 10^5^) were collected independently in three biological replicates and stored at −80 °C immediately after being frozen in liquid nitrogen. Total RNA was extracted from these samples, using TRIzol reagent (Invitrogen, Carlsbad, CA, USA). Then, using 1 μg RNA, a 20 μL cDNA system was synthesized according to the guidelines of the PrimeScript™ RT Reagent Kit with gDNA Eraser (TaKaRa, Otsu, Japan). qRT-PCR was performed on the Bio-Rad iCycler Real-Time PCR Detection System (Hercules, CA, USA). This was performed with three biological replicates, using TaKaRa SYBR Premix Ex Taq™ (Tli RNaseH Plus, TaKaRa, Otsu, Japan). The total reaction system consisted of 20 μL containing 10 μL of SYBR Premix (2×), 1 μL cDNA, 1 μL each of sense and antisense primer (10 μM), and 7 μL of ddH_2_O. The qRT-PCR program was structured as follows: an initial predenaturation phase at 95 °C for 1 min, followed by 40 cycles of denaturation at 95 °C for 10 s, annealing at 56 °C for 30 s, and a final extension phase at 72 °C for 30 s. We used Primer Premier 5.0 for primer pair design, and the NCBI Blast program was used to determine the specificity of all primers (see Appendix A). The β-actin gene (Mol013347) in *C. sinense* was used as an internal reference. Relative expression levels were calculated using the 2^−ΔΔCT^ method [104].

## 5. Conclusions

Within the scope of this study, a total of 68 *COL* genes were successfully identified across seven distinct orchid species, and their phylogenetic tree, gene structure, chromosome localization, collinearity, gene replication, selective pressure, cis-acting elements, expression patterns, and protein structure were characterized. Furthermore, we delved into the regulation of co-activity-associated transcription factors within *C. sinense*. Remarkably, we conducted an exploration of subcellular localization and performed regulatory analyses through the transient overexpression of two *CsiCOLs* in Cymbidium protoplasts. We plotted potential regulatory patterns concerning their impact on flowering time and flower development. Our findings offer valuable insights and a conceptual foundation for exploring the unique functions of orchid growth and development, which carry significant implications for contemporary orchid breeding practices.

## Figures and Tables

**Figure 1 ijms-24-16825-f001:**
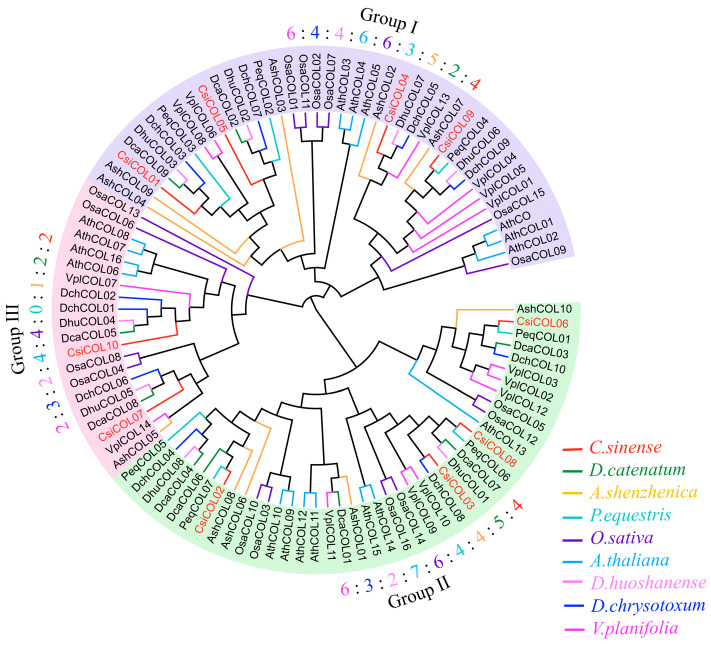
Molecular phylogenetic analysis of 101 *COL* proteins from *Arabidopsis. thaliana*, *Oryza. sativa*, *A. shenzhenica*, *D. catenatum*, *D. huoshanense*, *V. planifolia*, *D. chrysotoxum*, *C. sinense,* and *P. equestri.* Branches are colored according to the species color scheme on the bottom right.

**Figure 2 ijms-24-16825-f002:**
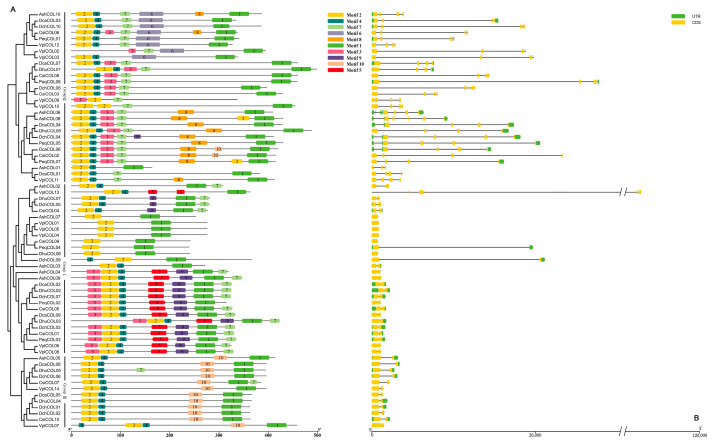
The conserved motifs and gene structure of the *COL* gene family in seven orchid species. (**A**) The conserved motifs of the *COL* gene family. (**B**) The gene structure of the *COL* gene family.

**Figure 3 ijms-24-16825-f003:**
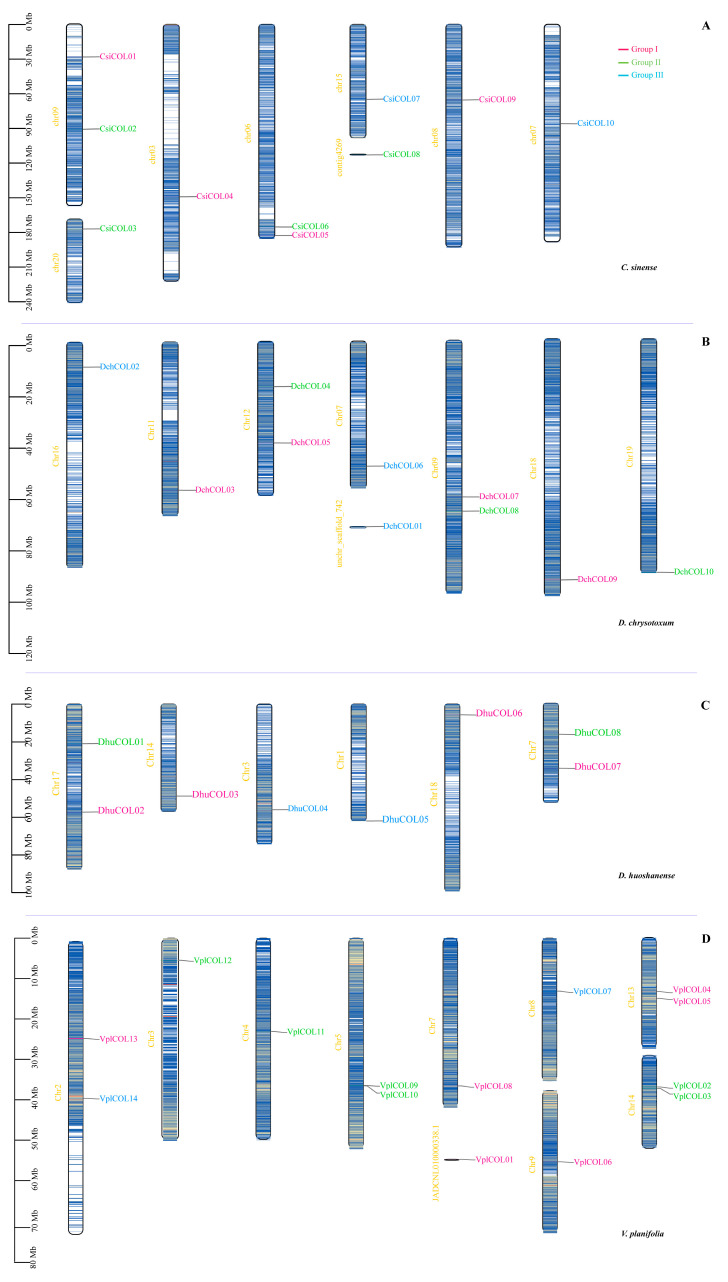
Chromosomal distribution of the *COL* genes in four orchids. Chromosome (Chr) names in yellow are on the left, and gene names are on the right. The scale on the left is in megabases (Mb). Gradient colors from red (high) to blue (low) indicate gene density in heat maps on orchid chromosomes by setting the estimated inheritance interval to 300 kb. Red indicates high gene density, and blue indicates low gene density. (**A**) *C. sinense*. (**B**) *D. chrysotoxum*. (**C**) *D. huoshanense*. (**D**) *V. planifolia*.

**Figure 4 ijms-24-16825-f004:**
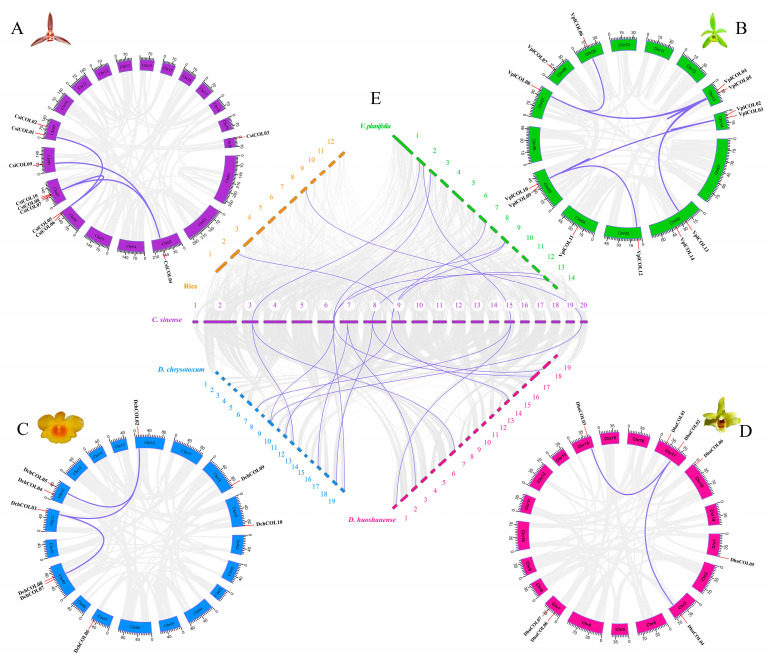
Chromosomal distributions of *COL* genes and schematic interchromosomal relationships. (**A**–**D**) Synteny analysis of four orchid genomes identifies segmental duplication pairs of *COL* subfamilies. Dark orchid and red colored lines indicate homologous pairs of COL genes and their corresponding chromosomal locations, respectively. (**E**) Intragenomic synteny between *C. sinense* (20 chromosomes) and each of rice (12 chromosomes), *D. chrysotoxum* (19 chromosomes), *D. huoshanense* (19 chromosomes), and *V. planifolia* (14 chromosomes).

**Figure 5 ijms-24-16825-f005:**
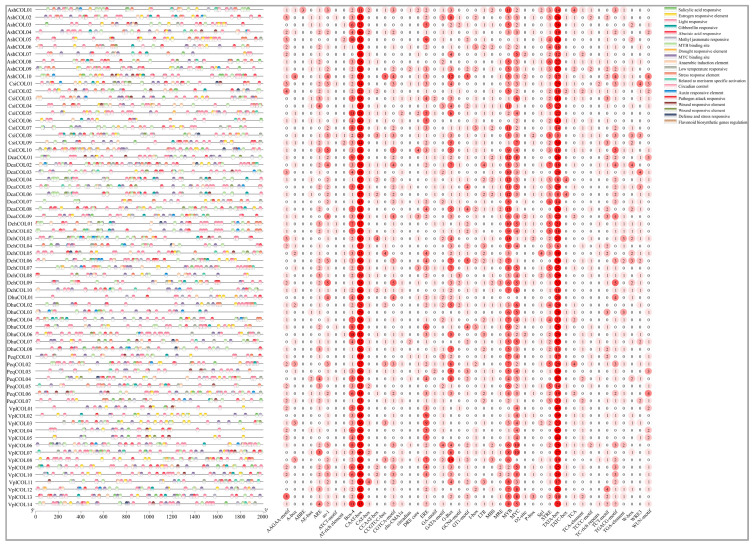
Cis-acting elements in the promoter regions of *COL* genes. Elements with similar regulatory functions are displayed in the same color. The number of each type of element is shown on the right.

**Figure 6 ijms-24-16825-f006:**
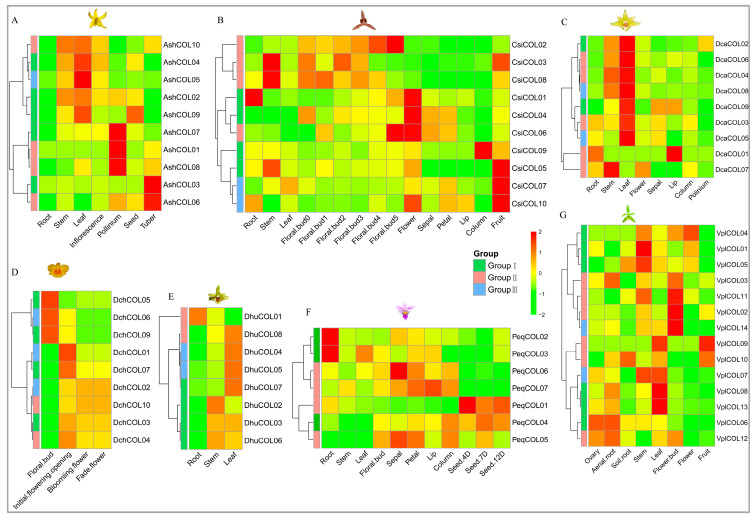
The expression profile of *COL* genes among different tissues in seven orchids. (**A**) *A. shenzhenica* and (**B**) *C. sinense*. Floral bud0, dormant lateral buds; floral bud1, 1−5 mm floral bud; floral bud2, 6−10 mm floral bud; floral bud3, 11−15 mm floral bud; floral bud4, 16−20 mm floral bud; floral bud5, blooming flower. (**C**) *D. catenatum*, (**D**) *D. chrysotoxum*, (**E**) *D. huoshanense*, (**F**) *P. equestri*, and (**G**) *V. planifolia*.

**Figure 7 ijms-24-16825-f007:**
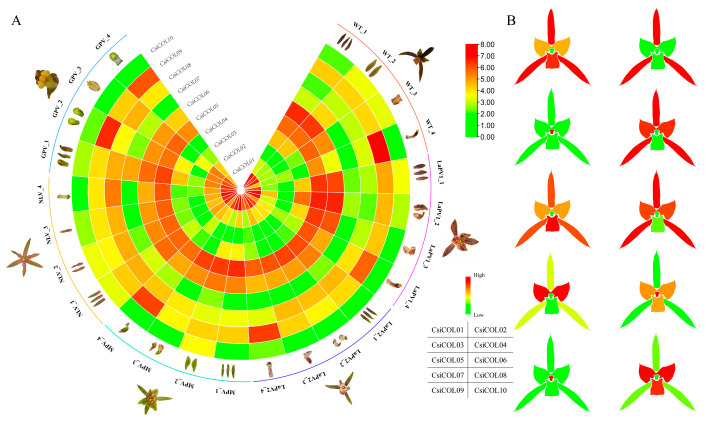
Gene expression patterns of *CsiCOL* genes in individual floral organs of different flower varieties. (**A**) Wild type (WT), gynostemium-like perianth variety (GPV), multi-perianth variety (MPV), labellum-like perianth variety (LaPV), and null-lip variety (NLV); 1 to 4 represent the individual floral organ sepal, petal, labellum, and column, respectively. (**B**) Cartoon heat map of tissue-specific expression of *CsiCOL* genes in *C. sinense* floral organs.

**Figure 8 ijms-24-16825-f008:**
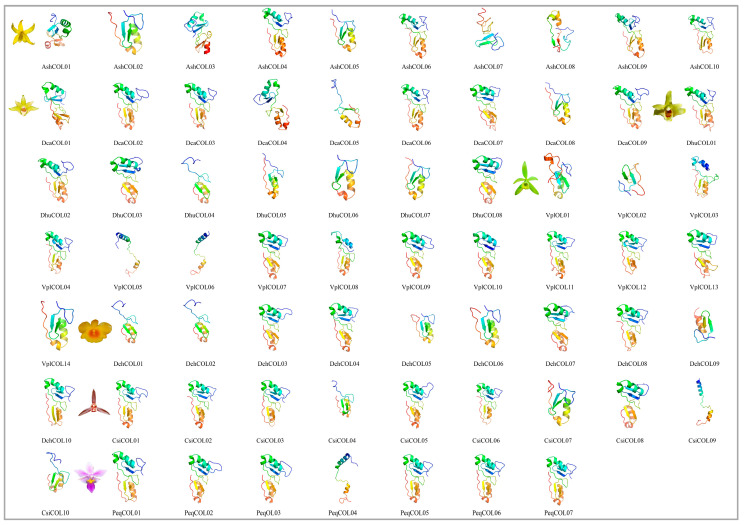
Protein tertiary structure of 68 *COL* genes from seven species of Orchidaceae. The tertiary structures are colored in rainbow order, representing the N-to-C terminuses.

**Figure 9 ijms-24-16825-f009:**
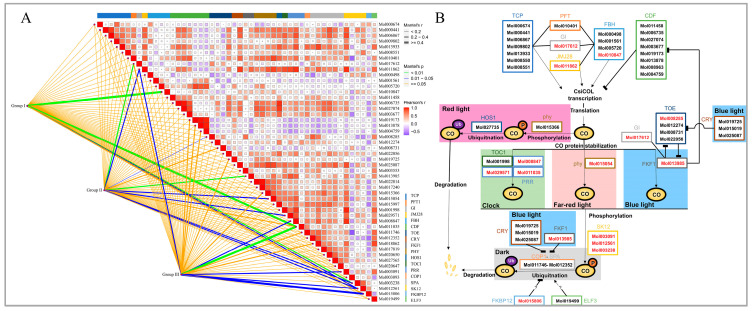
Representative regulatory network for *CsiCOLs* and *CO* regulatory genes. (**A**) Correlations of expression patterns between *CsiCOLs* and other transcription factors. (**B**) A model for *CsiCOLs* and *CO* regulators identified by their sequence homology to Arabidopsis in *C. sinense*.

**Figure 10 ijms-24-16825-f010:**
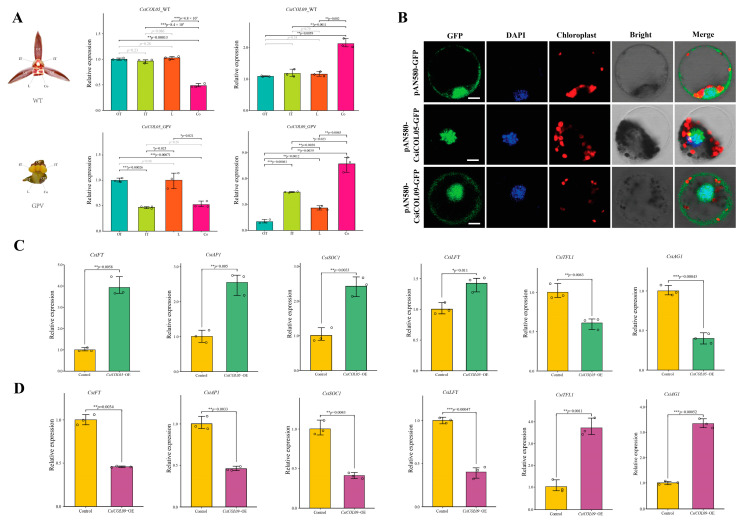
Subcellular localization and expression patterns of genes involved in flower development in response to the induction of *CsiCOLs*. (**A**) RT-qPCR analysis of *CsiCOL05* and *CsiCOl09* genes at different floral organs of standard and GPV variety *C. sinense*. (**B**) Subcellular localization of *CsiCOL05* and *CsiCOL09* protein in *C. sinense* protoplasts. Bar = 10 μm. (**C**,**D**) Relative expression levels of *CsiFT*, *CsiTFL1*, *CsiAP1*, *CsiSOC1*, *CsiLFY*, and *CsiAG1* genes in response to *CsiCOL05* and *CsiCOL09* induction by qRT-PCR, respectively. Significant differences analyzed between the *CsiCOLs*-OE and the WT (pAN580-GFP vector) by Student’s *t*-test, using R (*** *p*  <  0.001, ** *p*  <  0.01, and * *p*  <  0.05).

**Figure 11 ijms-24-16825-f011:**
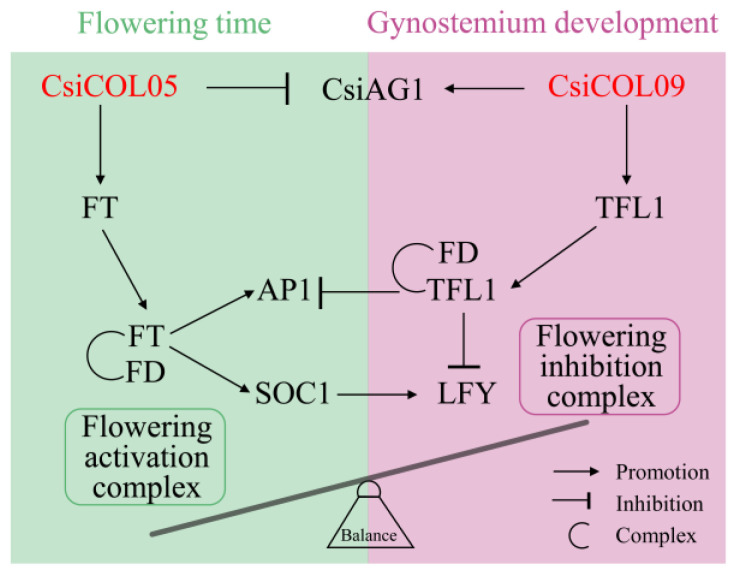
*CsiCOL05* and *CsiCOL09* schematics of potential regulatory patterns in *C. sinense*.

## Data Availability

Data are contained within the article and Appendix A.

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
