# Peer review of "Genome-Wide Identification, Expression, and Molecular Characterization of the CONSTANS-like Gene Family in Seven Orchid Species"

_ijms, 2023, doi:10.3390/ijms242316825_

Round 1
Reviewer 1 Report
Comments and Suggestions for Authors
The research topic presented in the reviewed manuscript entitled "Genome-wide identification, expression and molecular characterization of the CONSTANS-like gene family in seven orchids" written by YongLu Wei, JianPeng Jin, ZengYu Lin, ChuQiao Lu, Jie Gao, Jie Li, Qi Xie, Wei Zhu, Genfa Zhu and Fengxi Yang strictly concerns the genes that are involved in the flowering processes in orchids. The authors decided to study the CONSTANS-like (COL) gene family which are known for their importance in controlling flowering. The proper functioning of these genes is important for the flowers time regulation. There is no doubt that the proper development of flower elements plays the key role in processes connected with the seed production. As the object of the study, the authors chose seven plant species belonging to the large family of plants – Orchidaceae. Along with the Asteraceae family, Orchidaceae is the most abundant in species in the plant kingdom.
First of all I would like to suggest a minor change to the title of the paper. "Genome-wide identification, expression and molecular characterization of the CONSTANS-like gene family in seven orchid species" will reflect the contents of the paper better.
In the short (relatively) Introduction section, the authors gave proper and concise information about the Orchidaceae family, characteristics of the CONSTANS-LIKE (COL) genes, and justified the choice of the topic of the presented research.
The next section of the paper - 2. Results - is the most extensive part of the paper in which the authors present the performed experiments. No doubt, the results presented here are obtained through intensive studies. The results are described in detail with 10 figures. Each of them contains very extensive data connected with the main topic of the studies. Moreover, the large results are presented in a very smart and accessible form.
In the section “Discussion”, the authors briefly present the analysis of the results in comparison to published studies. Worthy of underlining is the addition of Figure 11, entitled “CsiCOL05 and CsiCOL09 schematics of potential regulatory patterns in C. sinense” which is a good supplement that summarizes the content of the paper.
The last section of the paper i.e., the Materials and Methods consists of information typical for such a section. The authors described all of the research methods that were used during performed research. The methods were well selected and used adequately to the planned goals of the research.
The conclusions in the section 5. are short, concise, and adequate to the results obtained by authors and are written in the proper way.
The paper contains a very abundant works cited (over one hundred entries).
The research that performed and presented in this paper is properly carried out. The results are also discussed in the proper way and sufficiently detailed. Therefore, I am convinced that this paper is very interesting, well written and suitable for publishing in the “International Journal of Molecular Sciences”.
Reviewer 2 Report
Comments and Suggestions for Authors
The present document develops the work made on genome-wide identification, expression and molecular characterization of the CONSTANS-like gene family in seven orchid.
The introduction is complete and clear, highlighting the gaps and presenting the problem.
The results are clearly presented and discussed abundantly.
Abstract and conclusion ok.
There are some english errors, and some form corrections to be performed, but is globally fine. You can find more details in the attached document. I've highlighted were there were the form problems and few other minor corrections.
Besides the comments I've give and the detailed corrections in the text, I think the research work have made progress on the knowledge on COLs genes family and inparticular in the family of Orchidaceae where few data is available in the literature. This knowledge will help to increase the flowering capacity in this family which will increase the possibilities for breeding, but also able to manage the flowering for ornamental purposes. As I recommend only minor corrections, I've no particular corrections to propose except the ones in the document attached. The conclusion matches and exposes the conclusions of the results and open to the future uses of the findings.

There are some expresions to be checked but globally is fine
Reviewer 3 Report
Comments and Suggestions for Authors
The authors of the manuscript “Genome-wide identification, expression and molecular characterization of the CONSTANS-like gene family in seven orchids” have identified 68 COL genes in the seven orchids genome and examined their properties, such as chromosome location, gene organization, cis-acting elements, protein-protein interactions, and gene expression pattern. They also overexpressed CsiCOL05 and CsiCOL09 genes in Cymbidium sinense protoplasts. There are numerous typographical errors, which the authors may introduce to reduce the plagiarism. The discussion in the results should be moved to the discussion.
L19: Cymbidium sinense instead of orchid.
L22: Pls avoid keywords from the title (suggestion- Cymbidium sinense; Dendrobium; Phalaenopsis).
L31: Vanilla (italic).
L35: Dendrobium (italic). Please italicize the botanical name throughout the text.
L36: Cymbidium (italic).
L39: environmental (typo).
L482: Escherichia coli (italic).
L61-62: Several reports are available on the functions of genes involved in orchid flowering. Please include the available literature.
L66, 67: C. sinense (italic).
L73-75: (Cymbidium sinense; Csi, Dendrobium catenatum; Dca, Dendrobium chrysotoxum; Dch, Dendrobium huoshanense; Dhu, Phalaenopsis equestri; Peq); one Apostasioideae (Apostasia shenzhenica; Ash) and one Vanilloideae (Vanilla planifolia; Vpl)
L95: C. sinense (space).
L110: . (delete).
L113-115: Provide space after the genus name, for example, O. sativa (Please correct it throughout the text).
L158: Please improve the figure quality.
L159: Chromosome (typo).
L168: chromosomal (typo).
L178: interchromosomal (typo).
L181: between (typo).
L188: respectively (typo).
L210: correlated (typo).
L213: meristematic (typo).
L256: A C‐terminal
L371: regulation (typo).
L372: Gynostemium (typo).
L378: participation (typo).
L390, 391: genostemium? gynostemium.
L392: described (typo).
L443, 444, 458, 473, 479, 498, 517 (correct the typos).
Comments on the Quality of English LanguageModerate editing of English language required
